# Perspectives of teachers of students with intellectual disabilities on the role of educational policies and systems in using Universal Design for Learning: A qualitative study

Sarah Khaled Alfawzan ⓘ*

Assistant Professor of Special Education, Department of Special Education, College of Education, King Faisal University, Al-Ahsa, Saudi Arabia

* salfozan@kfu.edu.sa

## Abstract

Universal Design for Learning is among the inclusive frameworks that has been incorporated in policies such as the "Every Student Succeeds Act of 2015" and the "Higher Education Opportunity Act of 2008". However, little is known about the factors that influence its use with students with intellectual disabilities, while their teachers play a critical role in understanding those factors. Currently, there is a paucity of research that has examined their perspectives in Saudi Arabia. Thus, the objective of this qualitative study was to explore teachers' perspectives on the role of educational policies and systems in using universal design for learning with their students. 14 teachers of students with intellectual disabilities were interviewed during the study. In the findings, the participants identified a number of factors that hinder its use, including: educational supervision; school management practices, and lack of access to the general curriculum, in addition to solutions to address them, such as providing legal/legislative support and funding for programs. The findings revealed unique experiences as well as commonalities among teachers, which are particularly important for policymakers, supervisors, principals, and teachers. Implications for future research, policy, and practice were presented.

## Introduction

Students with disabilities are being integrated into general education settings at unprecedented levels. However, most students with intellectual disabilities (ID) continue to be educated in separate settings throughout their school years [1]. When they do join general education students, it is often during non-academic learning experiences [2].

The Universal Design for Learning (UDL) framework has been recognized as an effective approach to facilitating the inclusion of students with IDs since 2006

**Data availability statement:** Data cannot be shared publicly because of [ the researcher guaranteed the privacy and confidentiality of all responses at every phase of the data collection process]. Data are available from the Scientific Research Ethics Committee at King Faisal University Institutional Data Access / Ethics Committee (contact via ialjreesh@kfu.edu.sa.) for researchers who meet the criteria for access to confidential data.

**Funding:** This study was funded by the Deanship of Scientific Research at King Faisal University, Saudi Arabia, grant number KFU251400. The funders had no role in study design, data collection and analysis, decision to publish, or preparation of the manuscript.

**Competing interests:** The authors have declared that no competing interests exist.

[2,3]. UDL aims to improve teaching and learning for all students by incorporating contemporary teaching methods, cognitive theory, and scientific insights into how individuals learn [4]. This framework encourages teachers to proactively consider the educational needs of students with and without disabilities. By addressing inflexible curricula that unintentionally create barriers to learning, UDL promotes the design of inclusive curricula and teaching methods that account for student diversity from the outset [5].

UDL is valued highly and supported in US education policy. It has also been incorporated into key legislative frameworks, including the Higher Education Opportunity Act of 2008, the National Education Technology Plan of 2016, and the Every Student Succeeds Act of 2015. As a framework, UDL guides the of design curricula and the development of instructional practices [6–9].

In 2014, the National Center for UDL also emphasized its role in addressing the barriers that special education teachers face when implementing inclusive educational practices to engage students with and without IDs. However, research on teachers' attitudes toward using UDL remains limited [10]. In recognizing this gap, the American Association on Intellectual and Developmental Disabilities (AAIDD) National Goals Conference in 2015 emphasized the need to assess the conditions and barriers that special education teachers encounter when implementing the UDL framework [10].

### Universal design for learning

UDL emerged from Universal Design (UD), which focuses on creating flexible spaces that accommodate all individuals regardless of their differences [11–13]. UDL also emerged from the disability movement, which recognizes that many of the barriers experienced by individuals with disabilities result from their interactions with the environment. In addressing these barriers, UDL shifts the focus from the individual model of disability to a social model, which emphasizes the presence of barriers within the physical and social environment [14,15].

Frolli [16] defines UDL as "an approach to instructional design that promotes the idea of producing physical environments and tools in the school system to enhance each student's experience" (p. 80). Kapil et al. [17] also note that it is "an educational framework that helps teachers reduce barriers and enhance learning opportunities for each student. It is designed to meet the diverse needs and abilities of all learners, eliminating unnecessary challenges in the learning journey" (p. 271). Researchers at the Center for Applied Special Technology have developed a set of three principles to align with the three learning networks. These principles emphasize the importance of using multiple methods of engagement, content delivery, and expression [18,19].

### UDL and promoting inclusion for students with ID

Educators and education professionals recognize significant differences among students in the classroom due to diversity in abilities, needs, and experiences [20]. Inclusive education seeks to integrate all students (with or without disabilities) into

their neighborhood schools and age-appropriate classrooms while providing the necessary support and reducing/removing barriers to prevent their exclusion. It also seeks to develop classroom design, programs, and activities to enable all students to learn and participate together, ensuring access to quality education by meeting their diverse needs [21].

UDL has gradually become an educational framework adopted widely to implement inclusive education strategies and achieve all students' participation [22]. It helps teachers reduce barriers and enhance learning opportunities for each student [17], and helps teachers plan inclusive lessons [23]. This framework has been praised and considered an effective means of facilitating the inclusion of students with ID since 2006 [2,3]. However, the National Center for UDL notes that special education teachers face tremendous pressure to implement inclusive practices in classrooms designed to provide access to the general curriculum for students with disabilities, including those with ID [22].

Yet some teachers resist including UDL in lesson planning, believing that implementing it involves more work and planning [24]. Additionally, some oversimplify UDL when complex differences exist [6]. Still other teachers struggle due to insufficient school administrative support, inadequate training for general education teachers to include students with disabilities in the classroom, and to provide training for all school staff [22]. This necessitates curriculum changes [15]. Furthermore, excluding students with disabilities from lesson planning undermines UDL as a truly inclusive framework for promoting inclusion [8]. Rao et al. [2] noted that current research on UDL implementation for students with ID focuses on students with other disabilities or their non-disabled peers, resulting in a diverse and fragmented understanding of UDL-based interventions and curriculum for students with ID.

## Education of students with ID in the Kingdom of Saudi Arabia (KSA)

The education sector in KSA has received substantial attention and financial investment, making it one of the countries committed to supporting individuals with disabilities [25,26]. Schools provide free special education services to students with disabilities to both citizens and residents in various settings, depending on the type and severity of disability [27]. For students with ID, KSA offers two types of educational environments: intellectual education institutes and integration programs. The former separates students with ID from those without disabilities and is generally reserved for those with severe disabilities, multiple disabilities, or autism. The other educational option is integration programs [28–30]. Schools supporting integration programs are responsible for providing all necessary educational and equipment materials and facilities to ensure students with ID receive similar educational opportunities [31].

## UDL for students with ID in the KSA

KSA has increasingly prioritized education for individuals with IDs, in line with global and local laws and regulations. The Education Strategy 2016–2020 highlighted the importance of ensuring quality and equitable education for all in the least restrictive environments [32]. The Saudi Ministry of Education has implemented several policies, such as Article (73) of the Regulations for Special Education Institutes and Programs, which mandates that students with disabilities can participate in classroom and extracurricular activities inside and outside of school [31]. Additionally, the policy promotes inclusive attitudes among general education classroom teachers and emphasizes the role of educational administrations in maintaining equal access for all students [32].

As part of these efforts and services, six public schools were built in Riyadh in 2016 to promote the integration of students with ID and establish an inclusive education project, with the potential to be implemented across the Kingdom [33]. However, educational services for students with ID in KSA remain in the development stage, compared to countries like the United States, where students with ID are often integrated into inclusive settings by special education teachers [33,34]. Furthermore, modern educational frameworks such as UDL have yet to be adopted across KSA. Few studies related to students with IDs have examined its application in schools in the KSA [34,35], although studies have confirmed its positive results in integrating students with non-disabled peers in the same classroom [35].

Still, several challenges hinder UDL's implementation in KSA. Many special education teachers and students are unfamiliar with UDL principles [34], and some struggle to understand how to effectively implement it [35]. This problem may be due to insufficient training and collaboration between general education teachers and special education teachers [36], as well as a lack of appropriate policies and guidelines for implementing the framework [34].

Educational policies and systems play a significant role in bringing about change, approving programs, and influencing student learning [37]. Inclusive policies and practices are one of the essential dimensions that inclusion experts have identified that promotes inclusive education within the classroom. In their Inclusion Index, Booth and Ainscow [38] emphasized the importance of creating a learning environment based upon values of justice, equality, and cooperation, which supports all students with diverse backgrounds and needs, particularly those with disabilities. The concept of "cultures" represents one of the three dimensions addressed by this index, and refers to establishing a positive school climate based upon a sense of belonging, forming supportive relationships, and removing psychological and social barriers that hinder participation and learning. In this context, the application of UDL can bring about a qualitative shift in the development of the learning and participation processes by providing multiple alternatives for presentation, expression, and participation that ensure equitable and effective learning opportunities for all students within a flexible and adaptable learning environment.

However, there is a lack of research on policies related to the use and implementation of inclusive frameworks in KSA. Therefore, understanding teachers' views towards UDL will address a gap in the research and contribute to the existing body of literature. Investigating the factors influencing UDL implementation, including practices, policies, laws, and funding, can provide insights for policy makers in developing appropriate legislation, laws, and support within schools and achieving success after transition. Additionally, using a qualitative approach may also help provide more insights and perspectives, as the effective use of the UDL framework depends on teachers' understanding of its objectives. However, there is a lack of relevant Saudi studies on teachers' perceptions of UDL, which limits efforts to expand UDL practice within the classroom.

## Materials and methods

### Methodological approach

On 13/10/2023, a qualitative study was conducted to engage deeply with the experiences of teachers of students with intellectual disabilities (ID), explore educational systems and policies' effect on their use of UDL with their students as experts in their own experiences [39], and elicit discussion on this topic in the context of Saudi culture. Using a phenomenological design as a methodological approach, accurate and rich data were obtained by identifying, understanding, and interpreting the participants' perspectives on challenges and potential solutions to address them from different relevant perspectives.

### Recruitment and research ethics

Purposive and snowball sampling were used to select participants. Purposive was used because my goal was to obtain variation in cities, schools, educational qualifications, and years of experience. Snowball sampling was employed because one teacher recommended other participants. The inclusion criteria were teachers who: (1) had more than five years of experience teaching students with ID; (2) taught at the intermediate level within the intellectual education programs attached to general education, and (3) were willing to participate in interviews. Male teachers were excluded from the study for practical reasons—Saudi culture encourages the segregation of males and females in schools, so the (female) researcher anticipated difficulty in both recruiting male respondents and developing an appropriate interview relationship with them, which could affect the quality of the data. In the end, 14 female teachers participated in this study. Table 1 shows the participants' characteristics. All teachers gave written consent to be interviewed. Ethical issues were addressed in this study, as Creswell and Poth [40] discussed, to maintain accuracy and adherence to ethical standards. The most

**Table 1. Characteristics of the participants.**

| Participant code | Educational qualification | Years of work experience |
|---|---|---|
| T1 | Bachelor of special education – intellectual disability | More than 15 |
| T2 | Master of education special | 10-less than 15 years |
| T3 | Bachelor of special education | 10-less than 15 years |
| T4 | Bachelor of special education | 10-less than 15 years |
| T5 | Bachelor of special education | More than 15 |
| T6 | Master of education special-inclusive education | 10-less than 15 years |
| T7 | Bachelor of special education | 5- Less than 10 years |
| T8 | Master of special education | 10-less than 15 years |
| T9 | Master of guidance and counseling | 10-less than 15 years |
| T10 | Bachelor of special education | 5- Less than 10 years |
| T11 | Master of special education | 5- Less than 10 years |
| T12 | Bachelor of special education | More than 15 |
| T13 | Master of special education | 10-less than 15 years |
| T14 | Master of special education | 5- Less than 10 years |

important of these concerns was ensuring the participants' anonymity by assigning them codes. They also confirmed that they were informed of the study and their rights, including the right to withdraw at any time. First, the researcher, an assistant professor of special education, obtained approval to perform the research from the Scientific Research Ethics Committee of University, after which she conducted the interviews.

## Interview procedure

Interviews were used as a method to collect data and acquire information to facilitate exploration and increase understanding of the issue [41]. The main focus of this study was to shed light on the participants' views on the political and organizational factors that influence the use of UDL with students with ID. A semi-structured individual interview was appropriate for its philosophy and purpose, as it contributed to examining the phenomenon from an individual perspective and gathering in-depth information [40]. Before the main study, three female teachers of students with ID were interviewed to develop interview guidelines (They were not included among the participants.), and asked them whether they had any suggestions or additions. Further, a qualitative study on this topic was used [42], which, together with their input, led to the development of the interview protocol (Table 2).

The interviews were conducted virtually via Zoom, as the participants were located in different cities. Before starting the recording, the researcher introduced herself (I introduced myself), explained the objectives of the study and its importance in improving the educational process and the access of students with ID to Inclusive education. The interviews took two months and were completed on 11/12/2023. It lasted between 35–77 minutes each. They were audio-recorded and transcribed verbatim.

## Data analysis procedure

The interviews were analyzed using a thematic analysis method. This approach allowed the challenges of implementing UDL and potential solutions to address them to be explored in-depth, as well as allowed the teachers' experiences and perspectives to be understood thoroughly. The first stage involved identifying and becoming familiar with the data. Each participant was assigned an identity code, the interviews were conducted and transcribed verbatim, and then analyzed manually. While the second stage involved coding and categorizing the data, the interviews were coded through an inductive reading of the transcripts. It includes moving from the part to the whole, arriving at the meanings that represent

**Table 2. Interview protocol.**

| Semi-structured Individual Interview Protocol | |
|---|---|
| Firstly, I would like to thank you for agreeing to participate in this study. Currently, I am studying the effect of educational policies and systems on the use of Universal Design for Learning for students with intellectual disabilities from the perspective of their teachers, given the importance of your role in your students' lives. Your answers to the interview questions will provide researchers with support to explore your perspectives and then make recommendations. Please note that your answers will be treated with complete confidentiality and will be used for scientific research purposes only. All audio recordings will be destroyed upon completion of their analysis, and no actual names will be mentioned anywhere in the study. Further, you are free to withdraw from participation at any time. | |
| 1 | Do you have prior knowledge and information about UDL, and how did you acquire it? |
| 2 | How would you describe your position and attitude toward using UDL? |
| 3 | From your own perspective, what are the most important benefits of using UDL for students with ID? |
| 4 | In your opinion, what are the main problems in the school/classroom that limit the use of UDL? Why? |
| 5 | Do you think that some teachers may not accept the use of UDL in their classrooms? Why? |
| 6 | From your own point of view, why is UDL not used in the Saudi educational system? |
| 7 | From your own perspective, what are your suggestions for using UDL in the classroom in the future to promote fair and inclusive teaching? |
| 8 | From your own point of view, what do you think might motivate teachers to include UDL in teaching and/or assessment? |
| 9 | From your own perspective as teachers, what are your most important needs from school administration to use UDL principles in the classroom? |
| 10 | What roles can stakeholders (e.g., the Ministry of Education) play to overcome the challenges of using UDL principles? |
| 11 | Before we end the interview, do you have any other additions that you would like to mention that could enhance understanding of this topic? |

the vision of individuals [43,44]. In the third stage, themes were identified. I examined the data categories to identify areas of similarity. After coding, the segments were grouped into themes, for example, political and organizational barriers were revealed during this stage. Eventually, some categories continued to form overarching themes, while others were consolidated further. Finally, the analysis revealed three overarching themes and seven sub-themes (Table 3).

## Trustworthiness

Credibility was enhanced through a triangulation strategy, involving data collection from multiple sources across several schools within the KSA and from 14 teachers of students with IDs. Dependability was also enhanced by providing a rich, in-depth description of the study methodology and procedures, ensuring transparency and accuracy. Confirmability was achieved through documentation of the procedures carried out throughout the study. Furthermore, transferability was considered, allowing the study findings to be applicable in similar situations [39,43]. To support validity, a detailed and accurate description of the research methods and procedures was provided, along with sufficient evidence in the form of quotes from interviews.

## Ethics statement

Approval from the Research Ethics Committee was obtained for this study from Scientific Research Ethics Committee at King Faisal University Ref. No. KFU-REC-2022-DEC-ETHICS424. The studies were conducted in accordance with Conducted in accordance with the principles set out in the Declaration of Helsinki, And the local legislation and institutional requirements. The participants provided their written informed consent to participate in this study.

**Table 3. Summary of the allocation of subthemes to main themes and the percentage of participants who discussed each subtheme.**

| Main themes | Subthemes | Percentage |
|---|---|---|
| 3.1. Attitudes toward UDL (This theme encompasses participants' opinions and perceptions regarding the concept of UDL and its application with their students.) | *3.1.1. Conceptualization* | 78% |
| | *3.1.2. Positive attitude* | 93% |
| 3.2. Challenging educational policies and systems in the use of UDL (This theme highlights the barriers and systemic constraints that hinder the implementation of UDL by teachers) | 3.2.1.*Educational supervision* | 64% |
| | 3.2.2.*School administration* | 43% |
| | 3.2.3.*Lack of access to the general curriculum* | 43% |
| 3.3. Solutions to address the challenges of educational policy and systems in the use of UDL (This theme focuses on the facilitators that support teachers' effective adoption of UDL) | *3.3.1. Legal/legislative support* | 71.4% |
| | *3.3.2. Funding programs for students with intellectual disabilities* | 64.2% |

## Findings

I identified three main themes: Attitudes toward UDL for students with ID; Challenging educational policies and systems in the use of UDL, and future solutions to address those challenges. These are illustrated below with representative quotes. Identifiers beginning with (T) refer to contributions from teachers (e.g., "T1" refers to the first participant interviewed).

### Attitudes toward UDL

14 participants revealed their opinions under two sub-themes: (1) Conceptualization, and (2) Positive attitude. *Conceptualization.* The participants demonstrated ambiguous and unclear perceptions of the UDL concept, and six participants showed overlap between UDL and the concept of inclusive education. However, in general, there was a relation to diversity in teaching methods and educational means, with an emphasis on using methods that suit personal abilities and preferences within general education or special education classrooms. With respect to the overlap with the concept of inclusive education, T2 indicated:

*"I believe that UDL is inclusive education, and it includes the inclusion of all students in one class. Currently, education departments are moving towards including more than one disability (autism, deafness, etc.) within the classroom, so that teachers can teach all groups, regardless of their disabilities and abilities."*

T3 also pointed to the relation between UDL and general education students' inclusion, saying:

"UDL is an educational framework for achieving success for all groups in an inclusive education environment, through which each group is taught in the most beneficial way for them. I thought that UDL was concerned with designing a curriculum in a way that suits the abilities of students with intellectual disabilities only and away from general education students, although some of their curricula are derived from general education. But when I read about it, I found that it includes their integration".

While six participants indicated that UDL involves adapting curricula and teaching methods to meet the diverse needs of all students within the general education classroom, T11 said:

"UDL aims to integrate students with disabilities with students without disabilities in the classroom, but we need to deliver information to the student in their own way and adapt the curriculum to suit their needs."

Five participants also indicated the need to create a context for implementing its principles. This includes preparing the school building for inclusive access for students with disabilities, ensuring that the educational tools used in the classroom

are easy for everyone to use, setting goals that are appropriate for diverse students, and adapting teaching methods. T6 stated:

> "UDL involves integrating students with disabilities into general education schools and requires that teaching methods and tools be inclusive of all."

With respect to the need to develop current curricula to implement UDL principles successfully, T5 stated:

> "UDL involves developing the curriculum to suit each student. Therefore, we need more developed curricula than current curricula, because students' abilities are not the same."

T9 indicated that UDL is related not only to the school context, but also extends to the community and requires family involvement:

> "UDL is holistic, relating to the student from an educational, social, and cultural perspective... It extends from school to home. Therefore, we need family involvement for good outcomes."

*Positive attitude.* When participants were asked their opinion on including these principles within the educational process, they agreed on their importance because of the diversity in the needs and inclinations of students with ID, in addition to the role that using multiple methods of integration, expression, and presentation plays in increasing these students' desire for education, achieving justice and fairness, and improving their academic performance, because of its belief in individual differences from the beginning. With respect to the use of the principle of multiple methods in integration/participation, ten participants indicated its role in increasing the students' interest in, and enthusiasm about the lesson, and giving everyone the opportunity to participate, in addition to developing thinking and establishing the pillars of the lesson in an enjoyable rather than boring way.

On the other hand, six participants pointed out its role in improving social experiences and achieving integration, as it helps increase the self-esteem, and thus the ability to confront society and escape isolation. Further, it helps the students learn some positive skills by imitating their peers, while the teacher's diversification in her use of strategies and activities motivates them to participate. With respect to this, T6 indicated:

> "We have a very shy student, and in order for the teacher to encourage her to participate, she asked her to write the answer on the small board and raise it for her colleagues to see, as her method of participation is through writing and not speaking, so it is important to choose a method that suits my student, as this will also keep her away from bullying".

When we discussed the principle of multiple methods of expression and assessment, 13 participants indicated that this principle seeks to achieve justice and fairness among students with ID in response to the diversity of their abilities and interests. It also contributes to achieving success and thus increasing self-esteem. The findings of this principle contribute to providing feedback to the teacher to determine what she can modify in her teaching method. With respect to the reality of the process of assessing students with ID, and the importance of using more than one method to do so, T6 stated:

> "We are used to assessment being through tests only, but the new trend in the Ministry of Education indicates the use of more than one method in assessment, for example, research, projects, or presentations. This principle is consistent with the diversity of skills among students. Therefore. Some may have a low level in a certain method, but that does not mean that the student is a failure, but rather that this method of assessment does not suit him or his preferences".

The findings also showed the positive role of using the multiple-methods principle in presentation, as 12 participants agreed on the role it plays in enhancing learning and creating a Inclusive, positive, and stimulating educational environment for students, as it responds to the diversity of learning styles within the classroom, and the difference in students' tendencies and interests, in addition to the speed of conveying, consolidating, and generalizing information.

## Challenging educational policies and systems in the use of UDL

12 participants revealed their views on the barriers to using UDL within three sub-themes: (1) educational supervision; (2) school administration, and (3) lack of access to the general curriculum.

*Educational supervision.* Interviews revealed the shortcomings in special education supervisors' professional competencies related to evaluating teachers' teaching practices, as these supervisors lack knowledge about the UDL framework, insist instead on adhering to traditional educational frameworks and strategies that do not support inclusiveness. In this respect, T11 said:

"Supervisors have no idea about UDL, so they are unable to communicate it to teachers. In our city, there is only one supervisor and she has a traditional way of evaluating teachers. I want her to guide me and teach me the right thing, but unfortunately, that does not happen. Others think that I am better than the supervisor".

The findings also showed that teachers are not given the flexibility to change and improve current lesson planning and implementation methods, as they are restricted to specific templates in the design of the Individual Education Plan (IEP). This affects these teachers' practices inside the classroom adversely, and limits their creativity/desire to apply modern educational frameworks, including UDL, or design an IEP based upon each student's needs and interests. Remember, T10 said:

"There is an important matter, the teacher must be given space for creativity. Unfortunately, we are deprived of that. We are not evaluated on how creative we are in the lesson, but rather on the number of pages we print for the student!"

*School administration.* The findings showed a number of hindering practices related to school administration, which included its separation from teaching practices and its lack of interest in the quality of the educational process within the classroom. With respect to this, T11 said:

"Administration is separate from the methods and strategies we use, and does not delve deeply into our use of UDL. What matters most to Administration is that the teacher is just teaching".

On the one hand, in addition, the administration assigns its teachers a heavy teaching load every day, so that the teacher has multiple tasks yet lacks time to complete them, which limits her ability to use UDL principles within the classroom. On the other hand, the findings indicated that the education administration restricts the school principal's powers, which affected many aspects adversely, including narrowing the educational environment for students with ID, as it resulted in long routine procedures, and then delays in obtaining official approvals for field trips or opening new classes.

*Lack of access to the general curriculum.* The findings indicated the Ministry of Education's need to provide access to the general curriculum and develop the design of the current curricula for students with ID, as they are not suitable for applying the principles of UDL and meeting the diverse needs of students, despite their periodic adjustment. The participants also stated that they are obligated to teach all objectives within the unified (special) curriculum to all students despite their different needs. In this respect, T13 indicated:

"Before, we used to design the curriculum ourselves, with 12-15 lessons per semester. We used to challenge students' abilities to reach their best. Now, the Ministry of Education has designed special curricula for students with intellectual

disabilities, but they are very simple and do not suit people with mild disabilities. Imagine teaching only four lessons in three months! It is boring and tiring".

With respect to the systems in the field of education that limit the teacher's ability to make modifications to the curricula, T12 says:

"The teacher is restricted by a specific template, and is not allowed to be creative or go beyond the curriculum and what the Ministry of Education has set! I present some enrichment topics to the students as a result of my sense of responsibility toward them, but I do not have the authority to include them in the official curriculum".

## Solutions to address the challenges of educational policy and the systems in the use of UDL

12 participants revealed their views on policy and systems solutions to address barriers to using UDL within two subtopics: (1) Legal/legislative support, and (2) Funding student programs with ID.

*Legal/legislative support.* The Interviews indicated the positive effect that will result when decision-makers support the framework through its inclusion in educational policy and guides, which indicating its importance and the obligation to implement it. In this respect, T2 indicated:

"The first step is to include it in the guide. Otherwise, only a few will care about it. The individual educational plans were initially included in the guide without implementation, and gradually, they reached the implementation as we see them now. Since the decision makers did not include UDL in the systems, how will we develop? As a consequence, it must be approved by the ministry, so that the multidisciplinary team feels the importance of its implementation".

With respect to the effect of including it in educational systems and guides on its application by teachers within the school, T6 stated:

"If UDL is not included in the organizational guide, it will not be implemented. Teachers will say, 'Why should we implement it when we [are] not obligated? Why should I exhaust myself and design a whole new curriculum with new methods and assessments?' There is another matter. We must study the extent to which reality is suitable for implementing UDL".

*Funding programs for students with ID.* The findings indicated the importance of funding intellectual education programs and providing a budget allocated to implement UDL principles. The teachers emphasized that it is important to evaluate this budget periodically according to current needs, given that the budget for special education programs has been stable for a long while, and there is a need to increase it. In addition, the budget must be distributed according to each educational stage. For example, the budget for the secondary stage must be different from other stages because of its connection to the transition. The participants also indicated the need to allocate it to educational applications rather than allocating it to celebrate some occasions, as is happening currently. In this respect, T2 indicated:

*"The education budget must be evaluated periodically, as it has unfortunately been static for a long time, and a portion of it must be allocated to the universal design for learning, given the need for a specific budget to implement this framework".*

T9 also indicated:

*"Modern technologies and specialists in various educational fields must be provided. Teachers alone cannot provide all of this. Funding is needed from the ministry, as it benefits and facilitates both teachers and students. There are*

*teachers who are not proficient in using technology optimally. Providing the above contributes to developing the teacher before the student, thus implementing UDL principles and improving outcomes."*

The participants also stressed the need to increase the budget allocated specifically to students with ID, as it is considered insufficient. T3 pointed out:

"The budget for programs for students with intellectual disabilities is not high, and those in charge of Saudi education do not have enough interest in UDL, and I think if it is presented for implementation, they will demand a larger budget, and thus, this will put pressure on the ministry, unlike the talent programs, which are allocated a very high budget".

## Discussion

### The vagueness of the concept of UDL and the positive attitude toward its principles

The findings related to opinions on UDL were summarized by revealing its concept and the positive attitude towards its principles. Overall, the participants showed ambiguity in defining its concept. This result is consistent with what other studies have indicated with respect to teachers' poor familiarity with the principles of UDL, and the need to improve their knowledge of its meaning and methods of use [6,22,24,34,45]. The participants also showed overlap in the concept of UDL and inclusive education, as both share the removal of barriers that may prevent students from achieving success and accessing learning opportunities in regular classrooms within the local community [46–48]. In this respect, it cannot be said that there are two different paths that inclusive education and UDL take. UDL contributes to all students' success and guarantees inclusive education for their benefit. If inclusive education welcomes all students with their differences, UDL facilitates this process, and through it, the classroom environment becomes suitable for everyone.

The participants in this study agreed on the positive attitude towards using UDL principles, as it provides all students with the opportunity to learn, given its emphasis on diversity in educational methods and strategies, which meet the similarly diverse needs within the classroom. Several studies [35,49,50] have indicated to this effect on students with ID in teaching academic content, improving their participation in activities, increasing their interest in lessons and academic achievement, and making learning opportunities more attractive and appropriate. The findings of this study also revealed teachers' awareness of the importance of diversity in educational methods and strategies to meet the aspects that result from the difference in the way that the brain works. The Center for Applied Special Technology indicated to that each person is unique because of a complex and intertwined brain network affected by genetics and environmental interactions, which can be predicted and organized across the three brain networks (cognitive, emotional, and strategic) through UDL [19]. This indicates that teachers accept diversity within the classroom, recognize that students do not learn in a linear path, and that it is necessary to provide diverse practices to access and succeed in inclusive education. This is consistent with the literature's results, which indicated an increase in the level of knowledge of teachers of students with ID of the importance of using UDL principles during teaching [45], their passion for implementing it [22], and its positive results in integrating students with ID in the same classroom with their peers without disabilities [35]. Some participating teachers indicated the need to introduce changes to certain current practices, particularly with respect to curricula and family involvement in the educational process. This reflects their awareness and understanding of the importance of enhancing collaboration between various stakeholders, which helps expand the scope of the application of UDL principles within the school environment.

### Challenges in the use of UDL

The findings showed many challenges to UDL's implementation, including special education supervisors' lack of knowledge of this framework, and their insistence on adhering to traditional educational strategies, although these supervisors

are considered the reference from the teachers' perspective and those who evaluate their performance. In this respect, the literature indicate [51,52] that special education supervisors' importance is not limited to evaluating their programs, but rather extends to developing their reality, consistent with modern practices in the field of educating students with disabilities. The supervisor is considered one of the main pillars of the educational system because of his essential role in ensuring the quality of education for these students, and the need for their teachers to support, develop, and provide feedback for them, which contributes to enhancing their skills and meeting these students' unique needs [53,54]. Therefore, supervisors' lack of knowledge about the UDL framework, and their failure to encourage teachers to use it will limit teachers' awareness of these Inclusive practices and their desire to apply them in the classroom, as it suggests that they are unimportant and are merely additional work. These supervisors are required to represent the Ministry of Education and the aspirations of Vision 2030 and the initiatives it adopts, most notably the inclusive education journey, which requires the existence of educational systems that enhance inclusion and access to the general curriculum [55]. This indicated the need to increase these supervisors' knowledge, improve their practices, and provide them with professional competencies that support their role in improving the quality of the education for students with disabilities in less restrictive environments, and work on their teachers' continuous professional development.

In the same context, the findings showed another challenge that faces UDL's implementation, which is related to a number of administrative practices that may result from officials in the Ministry of Education and school principals' lack of knowledge and skills associated with inclusive leadership, and insufficient sensitivity to issues related to the education of diverse students, and what their teachers should be provided. Altamimi [56] reported that knowledgeable leaders engage actively in special education programs, with increased understanding of disability-related needs, as well as provide resources for effective educational practices, and participate in decisions about programs and services. However, the findings indicated a disconnect between school administration and educational practices within the classroom. This may be attributable to the negative attitudes of some toward inclusive education; as a result, students with ID are being integrated only in part currently, and without the administration and its staff's appropriate preparation for their diverse needs. It was clear from their responses that the teachers believe that administrative support is necessary to help implement the UDL framework to meet the learning needs of students with ID effectively, which the literature has confirmed [22,57]. Hence, it is necessary for school principals to participate and work as a team to improve their knowledge and skills. Accordingly, in-service training programmes and courses should be provided that make them aware of their roles.

In their experience, the participants indicated the need to provide access to the general curriculum, as students with disabilities face challenges in this because of the lack of flexible teaching practices, which can be addressed through UDL [57–59], as the Individuals with Disabilities Education Act emphasizes enhancing access to general education curricula for people with disabilities through universally designed technologies [9]. Although policies in the Kingdom tend to provide services to individuals with disabilities within an inclusive environment to the extent possible [60], current systems' procedures limit this, as teachers are required to teach the entire curriculum (objectives/textbooks/lesson plans) to everyone at the end of each semester, which constitutes a barrier to access to general education and inclusive education curricula [61,62]. Compared to implementing UDL principles, which requires considerable time and effort, completing the curriculum is a priority for all teachers, which limits its use. Therefore, it is important to determine how to make curricula Inclusive and individualized according to UDL [2]. This requires decision-makers to make changes at the legal and policy level to ensure access to general education customized according to the needs of students with ID. It is also necessary to make special arrangements to design curricula, as well as develop new methods and strategies that are included in lesson design based upon UDL from the beginning [11,63,64].

## Solutions to address the challenges in the use of UDL

We asked teachers about possible solutions to address these challenges, and found that most of them mentioned the essential role of incorporating UDL into educational policies and guides to mandate or promote its use. This finding is

consistent with what Alquraini and Rao [34] reported about the lack of appropriate policies and guidelines to implement the framework. This contradicts Anstead's [34] result, in which the participants indicated that they only wanted to have guidelines for implementing UDL as an option rather than as an administrative mandate. Advocates of the American Disabilities Education Act encourage the use of research-based strategies to support inclusion efforts for students with disabilities [65]. Advocates of the "Every Student Education Act" also embraced the UDL framework's use by supporting the definition, approving the framework, and promoting its use, as the promotion and encouragement of this framework on the part of its advocates is important to implement its strategies within schools [22].

The participants indicated the need for more funding overall, as well as the need to allocate an increased budget for students with ID to enable teachers to implement educational strategies that support UDL principles. Currently, schools are not equipped with the necessary basic tools and facilities, although the education sector in the Kingdom receives a large share of attention and economic support, and is considered one of the countries that pays great attention to individuals with disabilities [25,26]. This result is consistent with what Jimenez et al. [66] reported, that schools do not receive sufficient funds to meet the needs that ensure ready access to education for students with ID. The "Every Student Succeeds Act" has provided funding opportunities to integrate UDL into K-12 educational environments [9], which confirms the need to support policies that provide more innovative funding to encourage the use of UDL within schools [22,57].

### Theoretical and practical implications of the UDL

This study contributed to the literature by using a qualitative approach that explored in-depth the details of implementing a research-based educational framework, UDL, which has seen increasing global interest while it has declined domestically. Using this approach also helped provide in-depth perspectives from teachers on the political and organizational factors that influence its use and a better understanding of it. Further, with its findings and recommendations, this study provides stakeholders with important information about the barriers and facilitators to using the UDL framework. Therefore, the study serves as a reference in the field of using its principles and meeting the diverse needs of students with ID to implement inclusive education through inclusive practices, which contributes thereby to the realization of the initiatives of the Kingdom of Saudi Arabia's vision. Further, the information from this study may help leaders and supervisors develop the current supervision and leadership process for education, as well as plan for teachers' professional development related to the factors that influence the use of UDL and provide resources to support it. This contributes to the development of the educational process and affects educational outcomes positively. Then, the results may contribute to achieving the Ministry of Education's development goals and future visions in the field of educating students with ID through research-based frameworks and practices that help them achieve inclusive education and academic and social success.

### Conclusions

This qualitative study explored the views of teachers of students with ID on the political and organizational factors that influence the implementation of the UDL framework with their students in Saudi Arabia. The findings revealed teachers' positive attitudes toward using the framework's principles, although they discussed certain challenges in its implementation, including educational supervision, school management practices, and lack of access to the general curriculum. The study also revealed solutions to address these challenges: legal/legislative support and funding for programs for students with ID. The findings emphasize the need to provide supervisors with professional competencies that support their role in setting policies and mechanisms, and providing high-quality educational services in accordance with Inclusive practices. In addition, it is necessary to involve school principals in the integration process, and to raise their awareness of their roles consistent with issues of special/inclusive education leadership. Improvements should be made at the legal and political levels to ensure that UDL is adopted and used with students with ID, while more funding and access to general education should be provided as well.

## Limitations and future research

In this study, the sample size was small and limited to several cities within Saudi Arabia because the goal was not to collect data that could be generalized. Rather, the objective was to obtain in-depth insights and clarify the study topic's specificity. Future research could use quantitative methods to collect data on stakeholders' perceptions, in which researchers could take advantage of drawing on a larger sample of the population so that they can report more general findings. The Kingdom's unique cultural traditions are likely to have influenced the findings of this study significantly. Therefore, it is difficult to apply them to any other country. Further, because the study was limited to female teachers for cultural reasons, there may be differences in male teachers' perceptions. Accordingly, this should be a consideration for future research. In future, researchers could examine the experiences and perspectives of general education teachers, specialists, principals, and administrators on the topic of this study, and attempt to find a relation among the variables.

## Author contributions

**Conceptualization:** Sarah K. Alfawzan.

**Data curation:** Sarah K. Alfawzan.

**Formal analysis:** Sarah K. Alfawzan.

**Funding acquisition:** Sarah K. Alfawzan.

**Investigation:** Sarah K. Alfawzan.

**Methodology:** Sarah K. Alfawzan.

**Project administration:** Sarah K. Alfawzan.

**Resources:** Sarah K. Alfawzan.

**Software:** Sarah K. Alfawzan.

**Supervision:** Sarah K. Alfawzan.

**Validation:** Sarah K. Alfawzan.

**Visualization:** Sarah K. Alfawzan.

**Writing – original draft:** Sarah K. Alfawzan.

**Writing – review & editing:** Sarah K. Alfawzan.

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
