## [Decision Letter · Decision Letter 0]

25 Jul 2025

PONE-D-25-18495Perspectives of teachers of students with intellectual disabilities on the role of educational policies and systems in using universal design for learning: A qualitative studyPLOS ONE

Dear Dr. Alfawzan,

Thank you for submitting your manuscript to PLOS ONE. After careful consideration, we feel that it has merit but does not fully meet PLOS ONE’s publication criteria as it currently stands. Therefore, we invite you to submit a revised version of the manuscript that addresses the points raised during the review process.

1. "Table 3. Summary of the allocation of subthemes to main themes and the percentage of participants who discussed each subtheme" Where is percentage? 

2. Define your themes, particularly what they represent. This will help the reader see whether the subthemes and quotations lie within this representation or not. For example, what ideas are to be cosidered as "Attitude toward UDL"? 

3. In your thematization, Atitude may suggest teachers disposition or inclination to use or resist UDL. In here expected subthemes could be Positive attitude and interest towards using the method or resistance or appreaciation, and may be conceptualization. However, effect of using UDL principles should come under a diffeerent theme. Besides, you are not interested in evaluating the effects of the method, rather teachers perspectives. I suggest this section needs to be revisited.

4. the thmes and subthemes are few and question the holistic understanding of teachers perspectives who implement UDL. for example, did they not say anything about how the principle could be adapted to your context? 

5. The discussion section begins with effect of adopting UDL principles. If this need be emerge as a theme, it has to come at the end. this needs re-strcturing the "findings" section and discussion section correspondingly.

6. Writing: "Challenging in the use of UDL" line 445. is it to mean Challenges?

7. You examinied perspectives of teachers in adopting UDL framework. I wish there could be a theretical and practical implication of the findings. By theoretical implication, you may highlight any contribution to the theory or framework. similarly, with practical implication, you may highlight how it can be successfully implemented and ensure quality of learning.8. Please adress reasonable comments of the reviewers. 

We look forward to receiving your revised manuscript.

Kind regards,

Bekalu Tadesse Moges

Academic Editor

PLOS ONE

Journal Requirements:

3. In this instance it seems there may be acceptable restrictions in place that prevent the public sharing of your minimal data. However, in line with our goal of ensuring long-term data availability to all interested researchers, PLOS’ Data Policy states that authors cannot be the sole named individuals responsible for ensuring data access (http://journals.plos.org/plosone/s/data-availability#loc-acceptable-data-sharing-methods).

Reviewers' comments:

Reviewer's Responses to Questions

**Comments to the Author**

1. Is the manuscript technically sound, and do the data support the conclusions?

Reviewer #1: Yes

Reviewer #2: Partly

2. Has the statistical analysis been performed appropriately and rigorously? 

Reviewer #1: N/A

Reviewer #2: N/A

3. Have the authors made all data underlying the findings in their manuscript fully available?

Reviewer #1: Yes

Reviewer #2: No

4. Is the manuscript presented in an intelligible fashion and written in standard English?

Reviewer #1: Yes

Reviewer #2: Yes

5. Review Comments to the Author

Reviewer #1: 1. UDL research development

Please note that UDL research has significantly evolved since 2020. Earlier sources, although valid, should not be presented as representative of the current theoretical framework.

2. Inclusive cultures

When discussing practices and policies, it would be appropriate to also mention inclusive cultures, referring to the three dimensions promoted by the Index for Inclusion (Booth & Ainscow, 2002).

3. Use of first person

To maintain an academic writing style, it is recommended to avoid first-person formulations (e.g., in the “Methodological Approach” section).

4. Participant selection criteria

On what basis (besides gender) were participants selected? What does “relevant information” (line 180) mean in this context? The selection process, as formulated, may introduce a potential bias.

5. Generalization of accommodations

It would be helpful to make more explicit the principle of generalising special attention to all students in the class, not just those with ID: "what is essential for some becomes useful for all".

6. Article structure

Some references or sections currently found in the “Discussion” would be more appropriately placed in the “Introduction.”

Reviewer #2: Dear author,

Thank you for the opportunity to review your manuscript. I believe this was an important topic for you to research. Here is my feedback:

- Page 7, under methodological approach, you say "I conducted..." but then on page 8, you move to describing yourself as "the researcher" at lines 179 and 194. I would stay consistent - and I personally much prefer first person instead of "the researcher...."

- Page 9, line 206, you say "we interviewed..." Who is the we? Previously you were saying "I" so I got confused here.

- Table 3, page 11 - check your numbering. It looked inaccurate to me

- Page 13, UDL Concept - can you expand on their knowledge of UDL? You really only have one paragraph, and then a lengthy quote. That doesn't really help me understand their knowledge of UDL at all.

- Page 15, line 317, and throughout until the end - you use the word "confirm" a lot. Qualitative research doesn't seek to "confirm" anything. I would find a different word to use throughout.

- Page 16, what is the difference between supervision and administration?

- Page 16, "...which affected many aspects adversely" - explain this more in depth.

- Page 17, How doe the two lengthy quotations relate to UDL? I don't see the relationship there at all.

- Page 17, "The results indicated the positive effect..." In what way do the results indicate this? Be more clear

- Page 18, Funding programs...What is the relation in this paragraph between budget and UDL? This was unclear to me.

- Page 19, Quotation at the top of the page - how does this relate to UDL? Make a connection between budget and UDL.

- Pages 20-22, watch for use of the word "confirm"

- Page 22 - This might be personal preference, but I prefer "findings" instead of "results" for qualitative research

6. PLOS authors have the option to publish the peer review history of their article (what does this mean? ). If published, this will include your full peer review and any attached files.

**Do you want your identity to be public for this peer review?** For information about this choice, including consent withdrawal, please see our Privacy Policy .

Reviewer #1: **Yes: ** Laura Rusconi

Reviewer #2: **Yes: ** Laura K. Anderson

---

## [Author Response · Author response to Decision Letter 1]

5 Aug 2025

Respond to editor's comments

"Table 3. Summary of the allocation of subthemes to main themes and the percentage of participants who discussed each subtheme" Where is percentage?

The percentage has been added to the table.

Attitude toward UDL

Define your themes, particularly what they represent. This will help the reader see whether the subthemes and quotations lie within this representation or not. For example, what ideas are to be cosidered as "Attitude toward UDL"?

Themes are defined, specifically what they represent.

3.1. Attitudes toward UDL

(This theme encompasses participants’ opinions and perceptions regarding the concept of UDL and its application with their students)

3.2. Challenging educational policies and systems in the use of UDL

(This theme highlights the barriers and systemic constraints that hinder the implementation of UDL by teachers)

3.3. Solutions to address the challenges of educational policy and systems in the use of UDL

(This theme focuses on the facilitators that support teachers' effective adoption of UDL)

In your thematization, Atitude may suggest teachers disposition or inclination to use or resist UDL. In here expected subthemes could be Positive attitude and interest towards using the method or resistance or appreaciation, and may be conceptualization. However, effect of using UDL principles should come under a diffeerent theme. Besides, you are not interested in evaluating the effects of the method, rather teachers perspectives. I suggest this section needs to be revisited.

This section has been revised and the themes have been changed to:

Subthemes have been changed:

• conceptualization

• Positive attitude

the thmes and subthemes are few and question the holistic understanding of teachers perspectives who implement UDL. for example, did they not say anything about how the principle could be adapted to your context?

Quotes have been added regarding how the principle can be adapted to context.

The discussion section begins with effect of adopting UDL principles. If this need be emerge as a theme, it has to come at the end. this needs re-strcturing the "findings" section and discussion section correspondingly.

The theme has been changed to a more relevant and appropriate one for the arrangement of findings and discussion:

The vagueness of the concept of UDL and the positive attitude towards its principles

Writing: "Challenging in the use of UDL" line 445. is it to mean Challenges?

Yes, done.

You examinied perspectives of teachers in adopting UDL framework. I wish there could be a theretical and practical implication of the findings. By theoretical implication, you may highlight any contribution to the theory or framework. similarly, with practical implication, you may highlight how it can be successfully implemented and ensure quality of learning.

A section has been added highlighting the contribution to the theory or framework. Similarly, through practical implication, it highlights how it has been successfully applied and ensured quality learning.

Please adress reasonable comments of the reviewers.

done

r1

UDL research development

Please note that UDL research has significantly evolved since 2020. Earlier sources, although valid, should not be presented as representative of the current theoretical framework.

A number of references published after 2020 were consulted, including:

Reference Nos. 4, 6, 9, 17, 19, 40, 48

Two new references were also added:

(Kapil et al., 2024), (Anderson, 2022).

Inclusive cultures

When discussing practices and policies, it would be appropriate to also mention inclusive cultures, referring to the three dimensions promoted by the Index for Inclusion (Booth & Ainscow, 2002).

Done.

Added in the introduction

Use of first person

To maintain an academic writing style, it is recommended to avoid first-person formulations (e.g., in the “Methodological Approach” section).

Done.

Modified in the Methodological Approach section.

Participant selection criteria

On what basis (besides gender) were participants selected? What does “relevant information” (line 180) mean in this context? The selection process, as formulated, may introduce a potential bias.

The inclusion criteria were teachers who: (1) had more than five years of experience teaching students with ID; (2) taught at the intermediate level within the intellectual education programs attached to general education, and (3) were willing to participate in interviews. Male teachers were excluded from the study for practical reasons

The phrase "relevant information" has been removed.

Generalization of accommodations

It would be helpful to make more explicit the principle of generalising special attention to all students in the class, not just those with ID: "what is essential for some becomes useful for all".

in this study, I have intentionally foregrounded individuals with intellectual disabilities, given their notable underrepresentation in qualitative research examining the Universal Design for Learning (UDL) framework. As Rao et al. (2017) observed, existing research on UDL implementation rarely includes students with intellectual disabilities as direct participants. Instead, it predominantly reflects the perspectives of students with other disabilities or those without disabilities, resulting in a fragmented and heterogeneous knowledge base concerning effective UDL-informed curricula and interventions tailored for students with intellectual disabilities.

My study also underscores the importance of extending individualized attention to all students within the classroom environment—not exclusively to those with intellectual disabilities. This inclusive approach is reflected throughout the manuscript, including:

- Introduction: References 4, 5, 10

- UDL: References 11, 12, 13, 17

- UDL and the promotion of inclusion for students with ID: Reference 22

Article structure

Some references or sections currently found in the “Discussion” would be more appropriately placed in the “Introduction.”

This was done through the following references: 2-9-11-22-23-24-25-33-34

r2

- Page 7, under methodological approach, you say "I conducted..." but then on page 8, you move to describing yourself as "the researcher" at lines 179 and 194. I would stay consistent - and I personally much prefer first person instead of "the researcher...."

Done

Page 9, line 206, you say "we interviewed..." Who is the we? Previously you were saying "I" so I got confused here.

Done

Table 3, page 11 - check your numbering. It looked inaccurate to me

Done

Page 13, UDL Concept - can you expand on their knowledge of UDL? You really only have one paragraph, and then a lengthy quote. That doesn't really help me understand their knowledge of UDL at all.

Done

- Page 15, line 317, and throughout until the end - you use the word "confirm" a lot. Qualitative research doesn't seek to "confirm" anything. I would find a different word to use throughout.

Done

Page 16, what is the difference between supervision and administration?

- Administration: Refers to internal school management personnel, including the school principal and deputy principal. This body is responsible for the execution of school operations and overseeing instructional practices within the school.

- Supervision: Refers to external educational personnel appointed by the Ministry of Education. Their primary role is to visit schools, observe classroom instruction, and evaluate teaching performance. Supervision operates independently from the school’s administrative structure and often serves as a liaison between policy implementation and classroom practice.

P age 16, "...which affected many aspects adversely" - explain this more in depth.

Participants emphasized that the school principal functions primarily as an executive body with no autonomous authority to initiate changes or issue decisions without prior approval from senior officials at the Ministry of Education. This restricted role limits the ability to respond promptly to pedagogical innovations.

One notable example concerned teachers seeking to apply the UDL principle of “multiple means of representation and engagement” through experiential learning—such as organizing field trips that allow students to participate in practical, real-world applications of lesson content. In such cases, obtaining formal approval entails navigating prolonged bureaucratic procedures at the ministerial level, often taking several months. Given that such delays extend into the academic semester’s final stages, the opportunity for implementation may be lost entirely. Participants suggested that had the school principal possessed the authority to approve such initiatives independently, the process could be expedited significantly—potentially requiring only a few days.

Page 17, How doe the two lengthy quotations relate to UDL? I don't see the relationship there at all.

The first quote highlights systemic barriers related to curriculum policies that hinder equitable access to the general curriculum and contribute to the persistence of the private curriculum. Specifically, it points to the rigid requirement of implementing a unified curriculum with standardized objectives for all students, irrespective of their individual backgrounds and learning profiles. This requirement directly contradicts the core philosophy of Universal Design for Learning (UDL), which emphasizes the uniqueness of each learner and advocates for personalized instructional goals.

Moreover, participants noted that previous approaches adopted by the Ministry of Education were more flexible and better aligned with UDL principles. These earlier measures granted teachers the autonomy to tailor learning objectives according to students' abilities and needs. In contrast, under current regulations, uniform objectives and lesson plans are mandated for all students. Such limited and simplified objectives fail to meet the needs of students with advanced capabilities, thereby impeding their cognitive development and depriving them of intellectually stimulating challenges.

The second quote further illustrates the limitations imposed on curriculum modification. It emphasizes that teachers are required to implement a standardized private curriculum for all students, with little or no room for adjusting objectives based on learner diversity within the classroom. Any attempt to modify the official curriculum—such as documenting alternative learning goals in the daily preparation file—is viewed as a violation of current educational regulations. This strict adherence undermines differentiated instruction and stands in contrast to the UDL framework, which inherently rejects the notion of a one-size-fits-all curriculum.

Page 17, "The results indicated the positive effect..." In what way do the results indicate this? Be more clear

It was replaced.

Participant interviews revealed that the commitment of decision-makers to the framework—demonstrated through its inclusion in official regulatory guidelines, continuous emphasis on its application, and mechanisms for accountability—constitutes a critical factor in securing teachers' engagement with it. Additionally, supervisory follow-up plays a central role in reinforcing its implementation, ensuring that educators adhere to the framework in practice.

Page 18, Funding programs...What is the relation in this paragraph between budget and UDL? This was unclear to me.

Participants emphasized that effective implementation of Universal Design for Learning (UDL) principles necessitates an increase in the current educational budget. This is due to the demand for diverse technical and material resources that support multiple modes of presentation, engagement, and expression. Additional requirements such as field trips, reinforcement activities, and other inclusive educational experiences further compound these needs. Several participants noted that teachers, operating without sufficient institutional support, are often unable to meet these demands alone—with some even reporting that they personally subsidize such resources from their own salaries.

Page 19, Quotation at the top of the page - how does this relate to UDL? Make a connection between budget and UDL.

The previously cited quote has been substituted with one that explicitly connects budgeting considerations to the principles of Universal Design for Learning

Pages 20-22, watch for use of the word "confirm"

Done

Page 22 - This might be personal preference, but I prefer "findings" instead of "results" for qualitative research

Done

---

## [Decision Letter · Decision Letter 1]

15 Sep 2025

Perspectives of teachers of students with intellectual disabilities on the role of educational policies and systems in using universal design for learning: A qualitative study

PONE-D-25-18495R1

Dear Dr. Sarah K. Alfawzan,

We’re pleased to inform you that your manuscript has been judged scientifically suitable for publication and will be formally accepted for publication once it meets all outstanding technical requirements.

Kind regards,

Bekalu Tadesse Moges

Academic Editor

PLOS ONE

Additional Editor Comments (optional):

Reviewer #1:

Reviewer #2:

Reviewers' comments:

Reviewer's Responses to Questions

**Comments to the Author**

1. If the authors have adequately addressed your comments raised in a previous round of review and you feel that this manuscript is now acceptable for publication, you may indicate that here to bypass the “Comments to the Author” section, enter your conflict of interest statement in the “Confidential to Editor” section, and submit your "Accept" recommendation.

Reviewer #1: All comments have been addressed

Reviewer #2: All comments have been addressed

2. Is the manuscript technically sound, and do the data support the conclusions?

Reviewer #1: Yes

Reviewer #2: Yes

3. Has the statistical analysis been performed appropriately and rigorously? 

Reviewer #1: N/A

Reviewer #2: N/A

4. Have the authors made all data underlying the findings in their manuscript fully available?

Reviewer #1: Yes

Reviewer #2: No

5. Is the manuscript presented in an intelligible fashion and written in standard English?

Reviewer #1: Yes

Reviewer #2: Yes

6. Review Comments to the Author

Reviewer #1: I have no additional comments to report. The author has revised the manuscript according to the suggestions and in line with the contextual circumstances in which the research is being conducted.

Reviewer #2: Thank you for addressing my recommendations! I appreciate the work you put in to making the changes I suggested. I do not have further comments at this time

7. PLOS authors have the option to publish the peer review history of their article (what does this mean? ). If published, this will include your full peer review and any attached files.

**Do you want your identity to be public for this peer review?** For information about this choice, including consent withdrawal, please see our Privacy Policy .

Reviewer #1: **Yes: ** Laura Rusconi

Reviewer #2: No

---

## [Editor Report · Acceptance letter]

PONE-D-25-18495R1

PLOS ONE

Dear Dr. Alfawzan,

I'm pleased to inform you that your manuscript has been deemed suitable for publication in PLOS ONE. Congratulations! Your manuscript is now being handed over to our production team.

Kind regards,

on behalf of

Dr. Bekalu Tadesse Moges

Academic Editor

PLOS ONE